# Deep Multi-task Gaussian Processes for Survival Analysis with Competing Risks

**Ahmed M. Alaa**
Electrical Engineering Department
University of California, Los Angeles
ahmedmalaa@ucla.edu

**Mihaela van der Schaar**
Department of Engineering Science
University of Oxford
mihaela.vanderschaar@eng.ox.ac.uk

## Abstract

Designing optimal treatment plans for patients with comorbidities requires accurate *cause-specific* mortality prognosis. Motivated by the recent availability of linked electronic health records, we develop a nonparametric Bayesian model for survival analysis with competing risks, which can be used for jointly assessing a patient's risk of multiple (competing) adverse outcomes. The model views a patient's survival times with respect to the competing risks as the outputs of a *deep multi-task Gaussian process* (DMGP), the inputs to which are the patients' covariates. Unlike parametric survival analysis methods based on Cox and Weibull models, our model uses DMGPs to capture complex non-linear interactions between the patients' covariates and cause-specific survival times, thereby learning flexible patient-specific and cause-specific survival curves, all in a data-driven fashion without explicit parametric assumptions on the hazard rates. We propose a variational inference algorithm that is capable of learning the model parameters from time-to-event data while handling right censoring. Experiments on synthetic and real data show that our model outperforms the state-of-the-art survival models.

## 1 Introduction

Designing optimal treatment plans for elderly patients or patients with comorbidities is a challenging problem: the nature (and the appropriate level of invasiveness) of the best therapeutic intervention for a patient with a specific clinical risk depends on whether this patient suffers from, or is susceptible to other "competing risks" [1-3]. For instance, the decision on whether a diabetic patient who also has a renal disease should receive dialysis or a renal transplant must be based on a joint prognosis of diabetes-related complications and end-stage renal failure; overlooking the diabetes-related risks may lead to misguided therapeutic decisions [1]. The same problem arises in nephrology, where a typical patient's competing risks are peritonitis, death, kidney transplantation and transfer to haemodialysis [2]. An even more common encounter with competing risks realizes in oncology and cardiovascular medicine, where the risk of a cardiac disease may alter the decision on whether a cancer patient should undergo chemotherapy or a particular type of surgery [3]. Since conventional methods for survival analysis, such as the Kaplan-Meier method and standard Cox proportional hazards regression, are not equipped to handle competing risks, alternate variants of those methods that rely on cumulative incidence estimators have been proposed and used in clinical research [1-7].

According to the most recent data brief by the Office of National Coordinator (ONC)[1], electronic health records (EHRs) are currently deployed in more than 75% of hospitals in the United States [8]. The increasing availability of data in EHRs has stimulated a great deal of research efforts that used machine learning to conduct clinical risk prognosis and survival analysis. In particular,

various recent works have proposed novel methods for survival analysis based on Gaussian processes [9], "temporal" logistic regression [10], ranking [11], and deep neural networks [12]. All these works have were restricted to the conventional survival analysis problem in which there is only *one event* of interest rather than a set of *competing risks*. (A detailed overview of previous works is provided in Section 3.) The usage of machine learning to construct data-driven survival models for patients with comorbidities is an important step towards *precision medicine* [13].

**Contribution** In the light of the discussion above, we develop a nonparametric Bayesian model for survival analysis with competing risks using deep (multi-task) Gaussian processes (DMGPs) [15]. Our model relies on a novel conception of the competing risks problem as a *multi-task learning* problem; that is, we model the *cause-specific* survival times as the outputs of a random vector-valued function [14], the inputs to which are the patients' covariates. This allows us to learn a "shared representation" of the patients' survival times with respect to multiple related comorbidities. The proposed model is Bayesian: we assign a prior distribution over a space of vector-valued functions of the patients' covariates [16], and update the posterior distribution given a (potentially right-censored) time-to-event dataset. This process gives rise to patient-specific multivariate survival distributions, from which a patient-specific, cause-specific cumulative incidence function can be easily derived. Such a patient-specific cumulative incidence function serves as *actionable information*, based upon which clinicians can design *personalized* treatment plans. Unlike many existing parametric survival models, our model neither assumes a parametric form for the interactions between the covariates and the survival times, nor does it restrict the distribution of the survival times to a parametric model. Thus, it can flexibly describe non-proportional hazard rates with complex interactions between covariates and survival times, which are common in many diseases with heterogeneous phenotypes (such as cardiovascular diseases [2]). Inference of patient-specific posterior survival distribution is conducted via a variational Bayes algorithm; we use *inducing variables* to derive a variational lower bound on the marginal likelihood of the observed time-to-event data [17], which we maximize using the adaptive moment estimation algorithm [18]. We conduct a set of experiments on synthetic and real data showing that our model outperforms state-of-the-art survival models.

## 2   Preliminaries

We consider a dataset $\mathcal{D}$ comprising survival (time-to-event) data for $n$ subjects who have been followed up for a finite amount of time. Let $\mathcal{D} = \{\mathbf{X}_i, T_i, k_i\}_{i=1}^n$, where $\mathbf{X}_i \in \mathcal{X}$ is a $d$-dimensional vector of covariates associated with subject $i$, $T_i \in \mathbb{R}_+$ is the time until an event occurred, and $k_i \in \mathcal{K}$ is the type of event that occurred. The set $\mathcal{K} = \{\emptyset, 1, \ldots, K\}$ is a finite set of $K$ mutually exclusive, competing events that could occur to subject $i$, where $\emptyset$ corresponds to *right-censoring*.

For simplicity of exposition, we assume that only one event occurs for every patient; this corresponds, for instance, to the case when the events in $\mathcal{K}$ correspond to deaths due to different causes. This assumption does not simplify the problem, in fact it implies the *nonidentifiability* of the event times' distribution parameters [6, 7], which makes the problem more challenging. Figure 1 depicts a time-to-event dataset $\mathcal{D}$ with patients dying due to either cancer or cardiovascular diseases, or have their endpoints censored. Throughout this paper, we assume *independent censoring* [1-7], i.e. censoring times are independent of clinical outcomes.

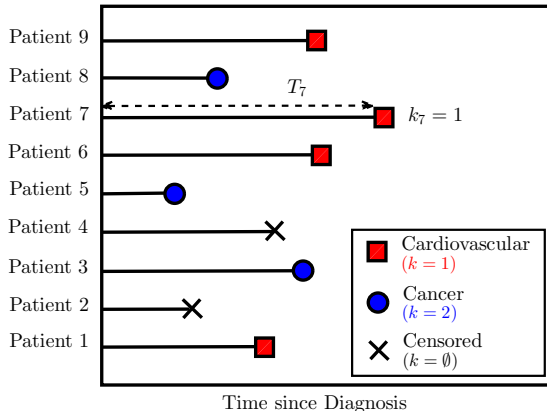

Figure 1: Depiction for the time-to-event data.

Define a multivariate random variable $\mathbf{T} = (T^1, \ldots, T^K)$, where $T^k, k \in \mathcal{K}$, denotes the *net survival time* with respect to event $k$, i.e. the survival time of the subject given that only event $k$ can occur. We assume that $\mathbf{T}$ is drawn from a conditional density function that depends on the sub-

ject's covariates. For every subject $i$, we only observe the occurrence time for the earliest event, i.e. $T_i = \min(T_i^1, \ldots, T_i^K)$ and $k_i = \arg\min_j T_i^j$.

The *cause-specific hazard function* $\lambda_k(t, \mathbf{X})$ represents the instantaneous risk of event $k$, and is formally defined as $\lambda_k(t, \mathbf{X}) = \lim_{dt \to 0} \frac{1}{dt} \mathbb{P}(t \leq T^k < t + dt, k \,|\, T^k \geq t, \mathbf{X})$ [6]. By the law of total probability, the overall hazard function is given by $\lambda(t, \mathbf{X}) = \sum_{k \in \mathcal{K}} \lambda_k(t, \mathbf{X})$. This leads to the notion of a *survival function* $S(t, \mathbf{X}) = \exp(\int_0^t \lambda(u, \mathbf{X}) du)$, which captures the probability of a subject surviving all types of risk events up to time $t$. The Cumulative Incidence Function (CIF), also known as the *subdistribution function* [2-7], is the probability of occurrence of a particular event in $k \in \mathcal{K}$ by time $t$, and is given by $F_k(t, \mathbf{X}) = \int_0^t \lambda_k(u, \mathbf{X}) \, S(u, \mathbf{X}) du$. Our main goal is to estimate the CIF function using the dataset $\mathcal{D}$; through these estimates, treatment plans can be set up for patients who suffer from comorbidities or are at risk of different types of diseases.

## 3 Survival Analysis using Deep Multi-task Gaussian Processes

We conduct patient-specific survival analysis by directly modeling the event times $\mathbf{T}$ as a function of the patients' covariates through the generative probabilistic model described hereunder.

**Deep Multi-task Gaussian Processes (DMGPs)** We assume that the net survival times for a patient with covariates $\mathbf{X}$ are generated via a (nonparametric) multi-output *random* function $g(.)$, i.e. $\mathbf{T} = g(\mathbf{X})$, and we use Gaussian processes to model $g(.)$. A simple model of the form $g(\mathbf{X}) = f(\mathbf{X}) + \epsilon$, with $f(.)$ being a Gaussian process and $\epsilon$ a Gaussian noise, would constrain $\mathbf{T}$ to have a symmetric Gaussian distribution with a restricted parametric form conditional on $\mathbf{X}$ [Sec. 2, 19]. This may not be a realistic construct for many settings in which the survival times display an asymmetric distribution (e.g. cancer survival times [2]). To that end, we model $g(.)$ as a Deep multi-task Gaussian Process (DMGP) [15]; a multi-layer cascade of vector-valued Gaussian processes that confer a greater representational power and produce outputs that are generally non-Gaussian. In particular, we assume that the net survival times $\mathbf{T}$ are generated via a DMGP with two layers as follows

$$\mathbf{T} = f_T(\mathbf{Z}) + \epsilon_T, \ \ \epsilon_T \sim \mathcal{N}(0, \sigma_T^2 \mathbf{I}),$$
$$\mathbf{Z} = f_Z(\mathbf{X}) + \epsilon_Z, \ \ \epsilon_Z \sim \mathcal{N}(0, \sigma_Z^2 \mathbf{I}), \tag{1}$$

where $\sigma_T$ and $\sigma_Z$ are the noise variances at the two layers, $f_T(.)$ and $f_Z(.)$ are two Gaussian processes with hyperparameters $\Theta_T$ and $\Theta_Z$ respectively, and $\mathbf{Z}$ is a hidden variable that the first layer passes to the second. Based on (1), we have that $g(\mathbf{X}) = f_T(f_Z(\mathbf{X}) + \epsilon_Z) + \epsilon_T$. The model in (1) resembles a neural network with two layers and an infinite number of hidden nodes in each layer, but with an output that can be described probabilistically in terms of a distribution. We assume that $f_T(.)$ has $K$ outputs, whereas $f_Z(.)$ has $Q$ outputs. The use of a Gaussian processes with two layers allows us to jointly represent complex survival distributions and complex interactions with the covariates in a data-driven fashion, without the need to assume a predefined non-linear transformation on the output space as it is the case in warped Gaussian processes [19-20].

A dataset $\mathcal{D}$ comprising $n$ i.i.d instances can be sampled from our model as follows:

$$f_Z \sim \mathcal{GP}(0, \mathbf{K}_{\Theta_Z}),$$
$$f_T \sim \mathcal{GP}(0, \mathbf{K}_{\Theta_T}),$$
$$\mathbf{Z}_i \sim \mathcal{N}(f_Z(\mathbf{X}_i), \sigma_Z^2 \mathbf{I}),$$
$$\mathbf{T}_i \sim \mathcal{N}(f_T(\mathbf{Z}_i), \sigma_T^2 \mathbf{I}),$$
$$T_i = \min(T_i^1, \ldots, T_i^K),$$

$i \in \{1, \ldots, n\}$, where $\mathbf{K}_\Theta$ is the Gaussian process kernel with hyperparameters $\Theta$.

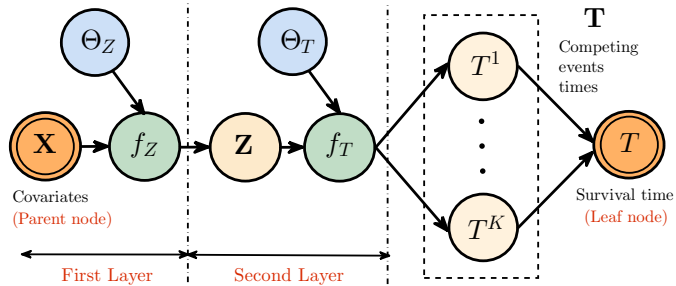

Figure 2: Graphical depiction for the probabilistic model.

Figure 2 provides a graphical depiction for our model (observable variables are in double-circled nodes); patient's covariates are the parent node; the survival time is the leaf node.

**Survival Analysis as a Multi-task Learning Problem**  As can be seen in (1), the cause-specific net survival times are viewed as the outputs of a vector-valued function $g(.)$. This casts the competing risks problem in a multi-task learning framework that allows finding a shared representation for the subjects' survival behavior with respect to multiple correlated comorbidities, such as renal failure, diabetes and cardiac diseases [1-3]. Such a shared representation is captured via the kernel functions for the two DMGP layers (i.e. $\mathbf{K}_{\Theta_Z}$ and $\mathbf{K}_{\Theta_T}$). For both layers, we assume that the kernels follow an *intrinsic coregionalization model* [14, 16], i.e.

$$\mathbf{K}_{\Theta_Z}(x, x') = \mathbf{A}_Z\, k_Z(x, x'), \quad \mathbf{K}_{\Theta_T}(x, x') = \mathbf{A}_T\, k_T(x, x'), \tag{2}$$

where $\mathbf{A}_Z \in \mathbb{R}_+^{Q \times Q}, \mathbf{A}_T \in \mathbb{R}_+^{K \times K}$ are positive semi-definite matrices, $k_Z(x, x')$ and $k_T(x, x')$ are *radial basis functions* with automatic relevance determination, i.e. $k_Z(x, x') = \exp\left(-\frac{1}{2}(x - x')^T \mathbf{R}_Z^{-1}(x - x')\right)$, $\mathbf{R}_Z = \text{diag}(\ell_{1,Z}^2, \ell_{2,Z}^2, \ldots, \ell_{d,Z}^2)$, with $\ell_{j,Z}$ being the *length scale* parameter of the $j^{th}$ feature ($k_T(x, x')$ can be defined similarly). Note that unlike regular Gaussian processes, DMGPs are less sensitive to the selection of the parametric form of the kernel functions [15]. This because the output of the first layer undergoes a transformation through a learned nonparametric function $f_Z(.)$, and hence the "overall smoothness" of the function $g(\mathbf{X})$ is governed by an "equivalent data-driven kernel" function describing the transformation $f_T(f_Z(.))$.

Our model adopts a Bayesian approach to multi-task learning: it posits a prior distribution on the multi-output function $g(\mathbf{X})$, and then conducts the survival analysis by updating the posterior distribution of the event times $\mathbb{P}(g(\mathbf{X}) \mid \mathcal{D}, \Theta_Z, \Theta_T)$ given the evidential data in the time-to-event dataset $\mathcal{D}$. The distribution $\mathbb{P}(g(\mathbf{X}) \mid \mathcal{D}, \Theta_Z, \Theta_T)$ does not commit to any predefined parametric form since it is depends on a random variable transformation through a nonparametric function $g(.)$. In Section 4, we propose an inference algorithm for computing the posterior distribution $\mathbb{P}(\mathbf{T} \mid \mathcal{D}, \mathbf{X}^*, \Theta_Z, \Theta_T)$ for a given out-of-sample subject with covariates $\mathbf{X}^*$. Once $\mathbb{P}(\mathbf{T} \mid \mathbf{X}^*, \mathcal{D})$ is computed, we can directly derive the CIF function $F_k(t, \mathbf{X}^*)$ for all events $k \in \mathcal{K}$ as explained in Section 2. A pictorial visualization of the survival analysis procedure assuming 2 competing risks is provided in Fig. 3.

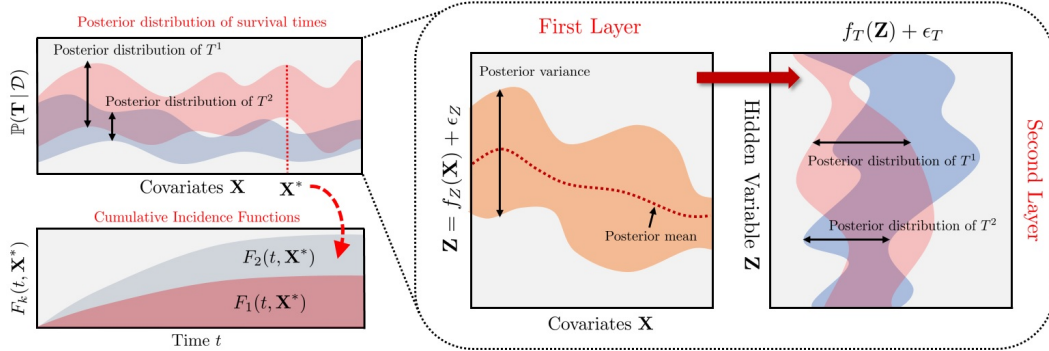

Figure 3: Pictorial depiction for survival analysis with 2 competing risks using deep multi-task Gaussian processes. The posterior distribution of $\mathbf{T}$ given $\mathcal{D}$ is displayed in the top left panel, and the corresponding cumulative incidence functions for a particular patient with covariates $\mathbf{X}^*$ is displayed in the bottom left panel. The posterior distributions on the two DMGP layers conditional on their inputs are depicted on the right panels.

**Related Works**  Standard survival modeling in the statistical and medical research literature is largely based on either the nonparametric Kaplan-Meier estimator [21], or the (parametric) Cox proportional hazard model [22]. The former is capable of learning flexible –and potentially non-proportional– survival curves but fails to incorporate patients' covariates, whereas the latter is capable of incorporating covariates, but is restricted to rigid parametric assumptions that impose proportional hazard curves. These limitations seems to have been inherited by various recently developed Bayesian nonparametric survival models. For instance, [24] develops a Bayesian survival model based on a Dirichlet prior, and [23] develops a model based on Gaussian latent fields, and proposes an inference algorithm that utilizes nested Laplace approximations; however, neither model incorporates the individual patient's covariates, and hence both are restricted to estimating a population-level survival curves which cannot inform personalized treatment plans. Contrarily, our model does not suffer from any such limitations since it learns patient-specific, nonparametric survival curves by adopting a Bayesian prior over a function space that takes the patients' covariates as an input.

A lot of interest has been recently devoted to the problem of survival analysis by the machine learning community. Recently developed survival models include random survival forests [26], deep exponential families [12], dependent logistic regressors [10], ranking algorithms [11], and semi-parametric Bayesian models based on Gaussian processes [9]. All of these methods are capable of incorporating the individual patient's covariates, but none of them has considered the problem of competing risks. The problem of survival analysis with competing risks has been only addressed through two classical parametric models: (1) the Fine-Gray model, which modifies the traditional proportional hazard model by direct transformation of the CIF [4], and (2) the threshold regression (multi-state) models, which directly model net survival times as the first hitting times of a stochastic process (e.g. Weiner process) [25]. Unlike our model, both models are limited by strong parametric assumptions on both the hazard rates, and the nature of the interactions between the patient covariates and the survival curves. These limitations have been slightly alleviated in [19], which uses a Gaussian process to model the interactions between survival times and covariates. However, this model assumes a Gaussian distribution as a basis for an accelerated failure time model, which is both unrealistic (since the distribution of survival times is often asymmetric), and also hinders the nonparametric modeling of survival curves. The model in [19] can be ameliorated via a warped Gaussian process that first transforms the survival times through a deterministic, monotonic non-linear function, and then applies Gaussian process regression on the transformed survival times [20], which would lead to more degrees of freedom in modeling the survival curves. Our model can be thought of as a generalization of a warped Gaussian process in which the deterministic non-linear transformation is replaced with another data-driven Gaussian process, which enables flexible non-parametric modeling of the survival curves. In Section 5, we demonstrate the superiority of our model via experiments on synthetic and real datasets.

## 4 Inference

As discussed in Section 3, conducting survival analysis requires computing the posterior probability density $d\mathbb{P}(\mathbf{T}^* \,|\, \mathcal{D}, \mathbf{X}^*, \Theta_Z, \Theta_T)$ for a given out-of-sample point $\mathbf{X}^*$ with $\mathbf{T}^* = g(\mathbf{X}^*)$. We follow an *empirical Bayes* approach for updating the posterior on $g(.)$. That is, we first tune the hyperparameters $\Theta_Z$ and $\Theta_T$ using the offline dataset $\mathcal{D}$, and then for any out-of-sample patient with covariates $\mathbf{X}^*$, we evaluate $d\mathbb{P}(\mathbf{T}^* \,|\, \mathcal{D}, \mathbf{X}^*, \Theta_Z, \Theta_T)$ by direct Monte Carlo sampling.

We calibrate the hyperparameters by maximizing the marginal likelihood $d\mathbb{P}(\mathcal{D} \,|\, \Theta_Z, \Theta_T)$. Note that for every subject $i$ in $\mathcal{D}$, we observe a "label" of the form $(T_i, k_i)$, indicating the type of event that occurred to the subject along with the time of its occurrence. Since $T_i$ is the smallest element in $\mathbf{T}$, then the label $(T_i, k_i)$ is *informative* of all the events (i.e. all the learning tasks) in $\mathcal{K}/\{k_i\}$; we know that $T_i^j \geq T_i, \forall j \in \mathcal{K}/\{k_i\}$. We also note that the subject's data may be right-censored, i.e. $k_i = \emptyset$, which implies that $T_i^j \geq T_i, \forall j \in \mathcal{K}$. Hence, the likelihood of the survival information in $\mathcal{D}$ is

$$d\mathbb{P}(\{\mathbf{X}_i, T_i, k_i\}_{i=1}^n \,|\, \Theta_Z, \Theta_T) \propto d\mathbb{P}(\{\mathcal{T}_i\}_{i=1}^n \,|\, \{\mathbf{X}_i\}_{i=1}^n, \Theta_Z, \Theta_T),$$

where $\mathcal{T}_i$ is a set of events given by

$$\mathcal{T}_i = \begin{cases} \{T_i^{k_i} = T_i, \{T_i^j \geq T_i\}_{j \in \mathcal{K}/\{k_i\}}\}, & k_i \neq \emptyset, \\ \{T_i^j \geq T_i\}_{j \in \mathcal{K}}, & k_i = \emptyset. \end{cases} \tag{3}$$

We can write the marginal likelihood in (3) as the conditional density by marginalizing over the conditional distribution of the hidden variable $\mathbf{Z}_i$ as follows

$$d\mathbb{P}(\{\mathcal{T}_i\}_{i=1}^n \,|\, \{\mathbf{X}_i\}_{i=1}^n, \Theta_Z, \Theta_T) = \int d\mathbb{P}(\{\mathcal{T}_i\}_{i=1}^n \,|\, \{\mathbf{Z}_i\}_{i=1}^n, \Theta_T) \, d\mathbb{P}(\{\mathbf{Z}_i\}_{i=1}^n \,|\, \{\mathbf{X}_i\}_{i=1}^n, \Theta_Z).$$

$$\tag{4}$$

Since the integral in (4) is intractable, we follow the variational inference scheme proposed in [15], where we tune the hyperparameters by maximizing the following variational bound on (4):

$$\mathcal{F} = \int_{\mathbf{Z}, f_z, f_T} \mathcal{Q} \cdot \log\left(\frac{d\mathbb{P}(\{\mathcal{T}_i\}_{i=1}^n, \{\mathbf{Z}_i\}_{i=1}^n, \{f_z(\mathbf{X}_i)\}_{i=1}^n, \{f_T(\mathbf{Z}_i)\}_{i=1}^n \,|\, \{\mathbf{X}_i\}_{i=1}^n, \Theta_Z, \Theta_T)}{\mathcal{Q}}\right),$$

where $\mathcal{Q}$ is a variational distribution, and $\mathcal{F} \leq \log\left(d\mathbb{P}(\{\mathcal{T}_i\}_{i=1}^n \,|\, \{\mathbf{X}_i\}_{i=1}^n, \Theta_Z, \Theta_T)\right)$. Since the event $\mathcal{T}_i$ happens with a probability that can be written in terms of a Gaussian density conditional on $f_Z$ and $f_T$, we can obtain a tractable version of the variational bound $\mathcal{F}$ by introducing a set of $M$ pseudo-inputs to the two layers of the DMGP, with corresponding function values $U^Z$ and $U^T$ at the first and second layers [15, 17], and setting the variational distribution to

$\mathcal{Q} = \mathbb{P}(f^T(\mathbf{Z}_i) \,|\, U^T, \mathbf{Z}_i)\, q(U^T)\, q(\mathbf{Z}_i)\, \mathbb{P}(f^Z(\mathbf{X}_i) \,|\, U^Z, \mathbf{X}_i)\, q(U^Z)$, where $q(\mathbf{Z}_i)$ is a Gaussian distribution, whereas $q(U^T)$ and $q(U^Z)$ are *free-form* variational distributions. Given these settings, the variational lower bound can be written as [Eq. 13, 15]

$$\mathcal{F} = \mathbb{E}\left[\log(d\mathbb{P}(\{\mathcal{T}_i\}_{i=1}^n \,|\, \{f^T(\mathbf{Z}_i)\}_{i=1}^n)) + \frac{\log(d\mathbb{P}(U^T))}{q(U^T)}\right]$$
$$+ \mathbb{E}\left[\log(d\mathbb{P}(\{\mathbf{Z}_i\}_{i=1}^n \,|\, \{f^Z(\mathbf{X}_i)\}_{i=1}^n)) + \frac{\log(d\mathbb{P}(U^Z))}{q(U^Z)}\right], \tag{5}$$

where the first expectation is taken with respect to $\mathbb{P}(f^T(\mathbf{Z}_i) \,|\, U^T, \mathbf{Z}_i)\, q(U^T)\, q(\mathbf{Z}_i)$ whereas the second is taken with respect to $\mathbb{P}(f^Z(\mathbf{X}_i) \,|\, U^Z, \mathbf{X}_i)\, q(U^Z)$. Since all the densities involved in (5) are Gaussian, $\mathcal{F}$ is tractable and can be written in closed-form. We use the adaptive moment estimation (ADAM) algorithm to optimize $\mathcal{F}$ with respect to $\Theta_T$ and $\Theta_Z$ [18].

# 5 Experiments

In this Section, we validate our model by conducting a set of experiments on both a synthetic survival model, and a real-world time-to-event dataset. In all experiments, we use the *cause-specific concordance index* ($C$-index), recently proposed in [27], as a performance metric. The cause-specific $C$-index quantifies the goodness of a model in ranking the subjects' survival times with respect to a particular cause/event based on their covariates: a higher $C$-index indicates a better performance. Formally, we define the (time-dependent) $C$-index for a cause $k \in \mathcal{K}$ as follows [Sec. 2.3, 27]

$$C_k(t) := \mathbb{P}(F_k(t, \mathbf{X}_i) > F_k(t, \mathbf{X}_j) \,|\, \{k_i = k\} \wedge \{T_i \leq t\} \wedge \{T_i < T_j \vee k_j \neq k\}), \tag{6}$$

where we have used the CIF $F_k(t, \mathbf{X})$ as a natural choice for the *prognostic score* in [Eq. (2.3), 27]. The $C$-index defined in (6) corresponds to the probability that, for a time horizon $t$, a particular survival analysis method prompts an assignment of CIF functions for subjects $i$ and $j$ that satisfy $F_k(t, \mathbf{X}_i) > F_k(t, \mathbf{X}_j)$, given that $k_i = k$, $T_i < T_j$, and that subject $i$ was not right-censored by time $t$. A high $C$-index for cause $k$ is achieved if the cause-specific CIF functions for a group of subjects who encounter event $k$ are likely to be "ordered" in accordance with the ordering of their realized survival times. In all experiments, we estimate the $C$-index for the survival analysis methods under consideration using the function `cindex` of the R-package `pec`[2] [Sec. 3, 27].

We run the algorithm in Section 4 with $Q = 3$ outputs for the first layer of the DMGP, and we use the default settings prescribed in [18] for the ADAM algorithm. We compare our model with four benchmarks: the Fine-Gray proportional subdistribution hazards model **(FG)** [4, 28], the accelerated failure time model using multi-task Gaussian processes **(MGP)** [19], the cause-specific Cox proportional hazards model **(Cox)** [27, 28], and the threshold-regression (multi-state) first-time hitting model with a multidimensional Wiener process **(THR)** [25]. The **MGP** benchmark is a special case of our model with 1 layer and a deterministic linear transformation of the survival times to Gaussian process outputs [Sec. 3, 19]. We run the **FG** and **Cox** benchmarks using the R libraries `cmprsk` and `survival`, whereas for the **THR** benchmark, we use the R-package `threg`[3].

## 5.1 Synthetic Data

The goal of this Section is to demonstrate the ability of our model to cope with highly heterogeneous patient cohorts; we demonstrate this by running experiments on two synthetic models with different types of interactions between survival times and covariates.

| Model A | Model B |
|---|---|
| $\mathbf{X}_i \sim \mathcal{N}(0, \mathbf{I})$, | $\mathbf{X}_i \sim \mathcal{N}(0, \mathbf{I})$, |
| $T_i^1 \sim \exp(\gamma_1^T \mathbf{X}_i)$, | $T_i^1 \sim \exp(\cosh(\gamma_1^T \mathbf{X}_i))$, |
| $T_i^2 \sim \exp(\gamma_2^T \mathbf{X}_i)$, | $T_i^2 \sim \exp(|\mathcal{N}(0,1) + \sinh(\gamma_2^T \mathbf{X}_i)|)$, |
| $T_i = \min\{T_i^1, T_i^2\}$, | $T_i = \min\{T_i^1, T_i^2\}$, |
| $k_i = \arg\min_{k \in \{1,2\}} T_i^k$, | $k_i = \arg\min_{k \in \{1,2\}} T_i^k$, |
| $i \in \{1, \ldots, n\}$. | $i \in \{1, \ldots, n\}$. |

In particular, we run experiments using the synthetic survival models **A** and **B** described above; the two models correspond to two patient cohorts that differ in terms of patients' heterogeneity. In

model **A**, we assume that survival times are exponentially distributed with a mean parameter that comprises a simple linear function of the covariates, whereas in model **B**, we assume that the survival distributions are not necessarily exponential, and that their parameters depend on the covariates in a nonlinear fashion through the $sinh$ and $cosh$ functions. Both models have two competing risks, i.e. $\mathcal{K} = \{\emptyset, 1, 2\}$, and for both models we assume that each patient has $d = 10$ covariates that are drawn from a standard normal distribution. The parameters $\gamma_1$ and $\gamma_2$ are 10-dimensional vectors, the elements of which are drawn independently from a uniform distribution. Given a draw of $\gamma_1$ and $\gamma_2$, a dataset $\mathcal{D}$ with $n$ subjects can be sampled using the models described above. We run 10,000 repeated experiments using each model, where in each experiment we draw a new $\gamma_1$, $\gamma_2$, and a dataset $\mathcal{D}$ with 1000 subjects; we divide $\mathcal{D}$ into 500 subjects for training and 500 subjects for out-of-sample testing. We compute the CIF function for the testing subjects via the different benchmarks, and based on those functions we evaluate the cause-specific $C$-index for time horizons $[1, 2.5, 7.5, 10]$. We average the $C$-indexes achieved by each benchmark over the 1000 experiments and report the mean value and the 95% confidence interval at each time horizon. In all experiments, we induce right-censoring on 100 subjects which we randomly pick from $\mathcal{D}$; for a subject $i$, right-censoring is induced by altering her survival time as follows: $T_i \leftarrow \text{uniform}(0, T_i)$.

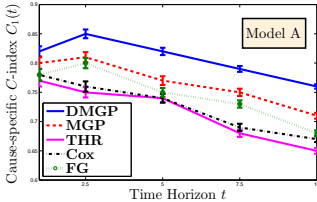 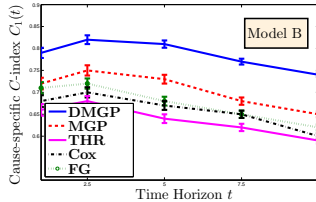 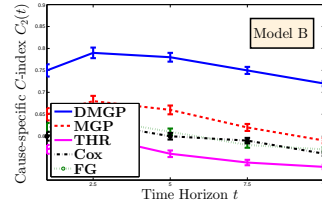

Figure 4: Results for model **A**.    Figure 5: Results for model **B**.    Figure 6: Results for model **B**.

Fig. 4, 5, and 6 depict the cause-specific $C$-indexes for all the survival methods under consideration when applied to the data generated by models **A** and **B** (error bars correspond to the 95% confidence intervals). As we can see, the DMGP model outperforms all other benchmarks for survival data generated by both models. For model **A**, we only depict $C_1(t)$ in Fig. 4 since the results on $C_2(t)$ are almost identical due to the symmetry of model **A** with respect to the two competing risks. Fig. 4 shows that, for all time horizons, the DMGP model already confers a gain in the $C$-index even when the data is generated by model **A**, which displays simple linear interactions between the covariates and the parameters of the survival time distribution. Fig. 5 and 6 show that the performance gains achieved by the DMGP are even larger under model **B** (for both $C_1(t)$ and $C_2(t)$). This is because model **B** displays a highly nonlinear relationship between covariates and survival times, and in addition, it assumes a complicated form for the distributions of the survival times, all of which are features that can be captured well by a DMGP but not by the other benchmarks which posit strict parametric assumptions. The superiority of DMGPs to MGPs shows the value of the extra representational power attained by adding multiple layers to conventional MGPs.

## 5.2   Real Data

More than 30 million patients in the U.S. are diagnosed with either cardiovascular disease (CVD) or cancer [1, 2, 29]. Mounting evidence suggests that CVD and cancer share a number of risk factors, and possess various biological similarities and (possible) interactions; in addition, many of the existing cancer therapies increase a patient's risk for CVD [2, 29]. Therefore, it is important that patients who are at risk of both cancer and CVD be provided with a joint prognosis of mortality due to the two competing diseases in order to properly manage therapeutic interventions. This is a challenging problem since CVD patient cohorts are very heterogeneous; CVD exhibits complex phenotypes for which mortality rates can vary as much as 10-fold among patients in the same phenotype [1, 2]. The goal of this Section is to investigate the ability of our model to accurately model survival of patients in such a highly heterogeneous cohort, with CVD and cancer as competing risks.

We conducted experiments on a real-world patient cohort extracted from a publicly accessible dataset provided by the Surveillance, Epidemiology, and End Results Program [4] (SEER). The extracted cohort contains data on survival of breast cancer patients over the years from 1992-2007. The total number of subjects in the cohort is 61,050, with a follow-up period restricted to 10 years.

The mortality rate of the subjects within the 10-year follow-up period is 25.56%. We divided the mortality causes into: (1) death due to breast cancer (13.64%), (2) death due to CVD (4.62%), and (3) death due to other causes (7.3%), i.e. $\mathcal{K} = \{\emptyset, 1, 2, 3\}$. Every subject is associated with 20 covariates including: age, race, gender, morphology information (Lymphoma subtype, histological type, etc), diagnostic confirmation, therapy information (surgery, type of surgery, etc), tumor size and type, etc. We divide the dataset into training and testing sets, and report the $C$-index results obtained for all benchmarks via 10-fold cross-validation.

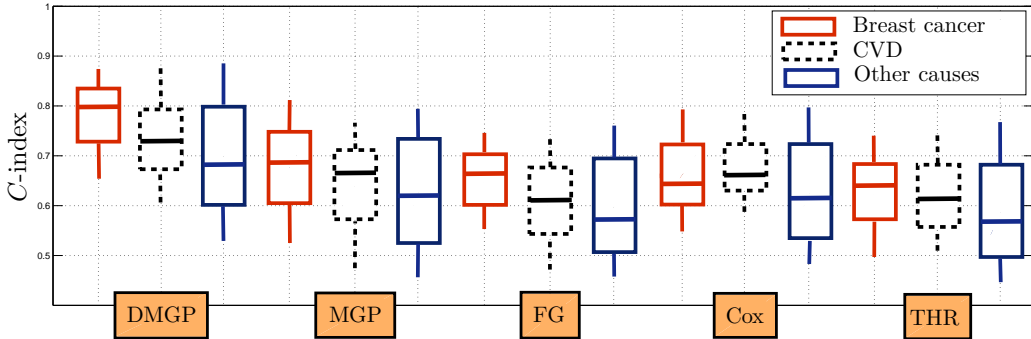

Figure 7: Boxplot for the cause-specific $C$-indexes of various methods. The $x$-axis contains the methods' names, and with each method, 3 boxplots corresponding to the $C$-indexes for the different causes are provided.

Fig. 7 depicts boxplots for the 10-year survival $C$-indexes (i.e. $C_1(10), C_2(10)$ and $C_3(10)$) of all benchmarks for the 3 competing risks. With respect to predicting survival times due to "other causes", the gain provided by DMGPs is marginal. We believe that this due to the absence of the covariates that are predictive of mortality due to causes other than breast cancer and CVD in the SEER dataset. The median $C$-index of our model is larger than all other benchmarks for all causes. In terms of the median $C$-index, our model provides a significant improvement in predicting breast cancer survival times while maintaining a decent gain in the accuracy of predicting survival times of CVD as well. This implies that DMGPs, by virtue of our nonparametric multi-task learning formulation, are capable of accurately (and flexibly) capturing the "shared representation" of the two "correlated" risks of breast cancer and CVD as a function of their shared risk factors (hypertension, obesity, diabetes mellitus, age, etc). As expected, since CVD is a phenotype-rich disease, predictions of breast cancer survival are more accurate than those for CVD for all benchmarks.

The competing multi-task modeling benchmark, MGP, is inferior to our model as it restricts the survival times to an exponential-like parametric distribution (See [Eq. 13, 19]). Contrarily, our model allows for a nonparametric model of the survival curves, which appears to be crucial for modeling breast cancer survival. This is evident in the boxplots of the cause-specific Cox benchmark, which is the only benchmark that performs better on CVD than breast cancer. Since the Cox model is restricted to a proportional hazard model with parametric, non-crossing survival curves, its poor performance on predicting breast cancer survival suggests that breast cancer patients have crossing survival curves, which signals the need for a nonparametric survival model [9]. This explains the gain achieved by DMGPs as compared to MGPs (and all other benchmarks), which posit strong parametric assumptions on the patients' survival curves.

## 6    Discussion

The problem of survival analysis with competing risks has recently gained significant attention in the medical community due to the realization that many chronic diseases possess a shared biology. We have proposed a survival model for competing risks that hinges on a novel multi-task learning conception of cause-specific survival analysis. Our model is liberated from the traditional parametric restrictions imposed by previous models; it allows for nonparametric learning of patient-specific survival curves and their interactions with the patients' covariates. This is achieved by modeling the patients' cause-specific survival times as a function of the patients' covariates using deep multi-task Gaussian processes. Through the personalized actionable prognoses offered by our model, clinicians can design personalized treatment plans that (hopefully) save thousands of lives annually.

## Footnotes

[1]https://www.healthit.gov/sites/default/files/briefs/

[2]`https://cran.r-project.org/web/packages/pec/index.html`

[3]`https://cran.r-project.org/web/packages/threg/index.html`

[4]https://seer.cancer.gov/causespecific/

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
