[Reviews · NeurIPS 2017]

Reviewer 1



The paper presents a method for survival analysis using deep Gaussian processes for applications with competing risks (i.e. comorbidities which need to be considered in determining treatment) using a multi-task approach where the (right-censored) survival times for each risk are viewed as different task, benefiting from a shared intermediate representation. The paper is very clear and well-written. The originality of the paper is good. While this is a somewhat of a niche application area of machine learning, I suspect the paper will be of high interest to those working in that area, as well as being a useful practical application of (multi-task) deep Gaussian processes. I am not expert in deep Gaussian processes, but as far as I can see the paper is technically sound. The experimental validation is reasonably compelling (hopefully a longer journal paper with more experimental details will be forthcoming?). I like this paper very much; it is always good to see recent advances in machine learning being applied to important real world problems, especially non-standard applications such as survival analysis. Line 130 should be "it depends on" (omit "is")? Line 140 should be "These limitations seem to have..." (rather than "seems")? Line 261 the use of "are diagnosed" suggests a rate with the time period missing, perhaps "have been diagnosed" if this refers to the total number of US citizens with an existing CVD and cancer diagnoses?

Reviewer 2



The paper presents a 2-layer Gaussian process model to fit survival time with competing risks, extending the MGP model as in reference 19 by removing the exponential link and adding one more layer of Gaussian process. The nonparametric model estimates patients' specific survival curves. But compare to other parametric models, it cannot tell the effects of covariates. A variational Bayes algorithm is given so that the inference is scalable. The novel model introduced, as well as many relevant existing concepts and models, are well reviewed. Numerical examples are provided to illustrate and support the methodological contributions. Empirically, the model outperforms all benchmarks and explanations are convincing. My only concern is about prediction. As stated in Line 75 to 83, people often care about hazard functions, survival functions or cumulative incidence functions. For the model in this paper, are all these functions in a closed form, or to be estimated by Monte Carlo method? The paper can be further improved if the authors give a comparison of these functions among different models.

Reviewer 3



The paper applies the deep multi-task GP for competing risk survival analysis. Experimental results on synthetic and real data comparing with four other methods are convincing. I have the following questions/comments 1. In line 176, it is not clear to me what "direct Monte Carlo sampling" entails here, since this involves the posterior (conditioned on data). 2. In line 195, what does "free-form" distribution mean -- perhaps a mixture of deltas? 3. I suggest to tabluate results for give in Figures 4 to 6, so that we can clearly see the variance of the results. 4. The conclusion from lines 289 to line 292 seems rather too strong without further investigation into the the learnt hyperparametsers and posterior. For example, can you explicitly show in the paper the "shared representation" that is mentioned? 5. The authors should include comparisons the Bayesian WGP [L\'{a}zaro-Gredilla, 2012].